# Risk Factors for the Development of Psoriasis

**DOI:** 10.3390/ijms20184347

**Published:** 2019-09-05

**Authors:** Koji Kamiya, Megumi Kishimoto, Junichi Sugai, Mayumi Komine, Mamitaro Ohtsuki

**Affiliations:** Department of Dermatology, Jichi Medical University, 3311-1 Yakushiji, Shimotsuke, Tochigi 329-0498, Japan (M.Ki.) (J.S.), (M.K.) (M.O.)

**Keywords:** psoriasis, risk factor, extrinsic risk factor, intrinsic risk factor, onset, exacerbation

## Abstract

Psoriasis is an immune-mediated genetic skin disease. The underlying pathomechanisms involve complex interaction between the innate and adaptive immune system. T cells interact with dendritic cells, macrophages, and keratinocytes, which can be mediated by their secreted cytokines. In the past decade, biologics targeting tumor necrosis factor-α, interleukin (IL)-23, and IL-17 have been developed and approved for the treatment of psoriasis. These biologics have dramatically changed the treatment and management of psoriasis. In contrast, various triggering factors can elicit the disease in genetically predisposed individuals. Recent studies suggest that the exacerbation of psoriasis can lead to systemic inflammation and cardiovascular comorbidity. In addition, psoriasis may be associated with other auto-inflammatory and auto-immune diseases. In this review, we summarize the risk factors, which can be divided into two groups (namely, extrinsic and intrinsic risk factors), responsible for the onset and exacerbation of psoriasis in order to facilitate its prevention.

## 1. Introduction

Psoriasis is a chronic inflammatory skin disease characterized by sharply demarcated erythematous plaques with whitish scale [1,2]. Psoriasis is one of the most frequent chronic inflammatory skin diseases. The prevalence of psoriasis varies with the country, and psoriasis can appear at any age [3,4], suggesting that ethnicity, genetic background, and environmental factors affect the onset of psoriasis. Genetic factors play a significant role in the pathogenesis of psoriasis. Psoriasis susceptibility 1 (PSORS1), which lies within an approximately 220 kb segment of the major histocompatibility complex on chromosome 6p21, is a major susceptibility locus for psoriasis [5,6,7]. HLA-Cw6 is the susceptibility allele within PSORS1 [8]; it is associated with early onset and severe and unstable disease [8,9]. In genetically predisposed individuals, various triggering factors can elicit the disease. In past surveys from 1982 to 2012, the exacerbating factors for the Japanese population were observed to be stress (6.4% to 16.6%), seasonal factors (9.7% to 13.3%), infection (3.5% to 8.3%), sun exposure (1.3% to 3.5%), and β-blockers (0.9% to 2.3%) [10,11,12]. The comorbidities included hypertension (1.1% to 27.8%), diabetes mellitus (DM) (7.0% to 13.9%), cardiovascular diseases (4.2% to 8.1%), and tonsillitis (3.5% to 5.4%) [10,11,12]. The risk factors for psoriasis can be divided into two groups, namely, extrinsic and intrinsic risk factors (Figure 1). In this review, we focus on each component of these groups and discuss their effects on the development of psoriasis.

## 2. Extrinsic Risk Factors

### 2.1. Mechanical Stress

In patients with psoriasis, skin lesions appear in uninvolved areas after various injuries [13,14,15,16]; this is known as the Koebner phenomenon. Radiotherapy, ultraviolet (UV) B, and even a slight skin irritation have been reported to trigger new lesions of psoriasis [17,18,19]. However, psoriatic lesions are not always observed in the uninvolved skin after injuries [20,21]. Type, site, depth, and degree of trauma may affect the pathogenesis of the Koebner phenomenon [20]. Under appropriate conditions, the Koebner phenomenon may occur, especially when there is dermal trauma with epidermal involvement. It is speculated that increased papillary dermis blood flow helps bring mediators that play a part in the pathogenesis of psoriasis [20]. However, the mechanisms underlying the Koebner phenomenon remain to be completely elucidated [20,21]. Nerve growth factor (NGF) is a neurotrophic factor that is expressed in both the nervous system and peripheral organs. NGF is thought to be associated with the Koebner phenomenon [22]. After a cutaneous trauma, in a developing psoriasis lesion, keratinocyte proliferation and up-regulation of NGF in basal keratinocytes are early events and precede epidermotropism of T lymphocytes [22]. In addition, NGF secreted by the psoriatic keratinocytes is functionally active. Notably, the keratinocytes of patients with psoriasis produce higher levels of NGF. This study suggests that NGF plays a critical role in the pathogenesis of psoriasis and that the regulatory role of NGF and its receptor system is functionally active in the early stage of developing lesions of psoriasis. Resident memory T cells (T_RM_) have been described as a non-circulating memory T cell subset that persists long-term in peripheral tissues; psoriasis is one of the T_RM_-mediated autoimmune inflammatory skin diseases [23]. Interestingly, psoriasis lesions could be triggered and sustained by skin-resident pathogenic T cells in the non-lesioned skin of psoriasis patients [24,25]. Activation of resident T cells is necessary and sufficient for the development of lesions in psoriasis [24]. A subpopulation of T cells infiltrating the epidermis during active disease turn into T_RM_ cells, and T_RM_ cells are retained in resolved psoriasis [26]. These cells establish a site-specific disease memory and are capable of producing cytokines that play a critical role in the pathogenesis of psoriasis [26]. These observations suggest that T_RM_ cells are key players not only in the recurrent lesions of psoriasis but also in the lesions of the Koebner phenomenon. Type 1 interferons (IFNs), such as IFN-α and IFN-β, have been suggested to play an indispensable role in initiating psoriasis during skin injury [27]. Skin injury rapidly induces IFN-β from keratinocytes and IFN-α from dermal plasmacytoid dendritic cells through distinct mechanisms [27]. Host antimicrobial peptide LL37 potentiates double-stranded RNA immune pathways and single-stranded RNA or DNA pathways in plasmacytoid dendritic cells. Production of type 1 IFNs induced by skin injury may explain the Koebner phenomenon.

### 2.2. Air Pollutants and Sun Exposure

The increase in air pollution over the years has had major effects on the human skin, and various air pollutants such as polycyclic aromatic hydrocarbons, volatile organic compounds, oxides, particulate matter, ozone, heavy metals, and UV damage the skin by inducing oxidative stress [28]. Cadmium is one of the air pollutants which affect the pathogenesis of psoriasis. Patients with severe psoriasis had higher blood cadmium when compared with the general population [29]. This study suggests that environmental exposure to cadmium may compromise immunity, and microenvironmental perturbation can predispose one to the worsening of psoriasis. The UV radiation that reaches the Earth’s surface is divided into two subtypes: more than 95% UVA (315–400 nm) and 1%–5% UVB (280–315 nm). In the past several decades, phototherapy has been widely used to treat psoriasis [30]. Both narrowband UVB (311 nm) and excimer laser (308 nm) are currently used as the first-line therapy for psoriasis, and psoralen UVA (PUVA) is also used as the second-line therapy with preference to refractory psoriatic plaques [30]. There is a subset of patients with severely photosensitive psoriasis in whom the condition is predominantly photodistributed and is severe in the summer months [31]. In this study, patients with photosensitive psoriasis showed striking female predominance, very low mean age of psoriasis onset, family history of psoriasis, a strong HLA-Cw*0602 association, and a rapid abnormal clinical response to broadband UVA, comprising erythema and/or scaling plaques [31]. A phenotypically distinct subset of psoriasis was characterized by histopathological analysis. In a certain group, psoriasis can develop after UV exposure.

### 2.3. Drugs

Drug-related psoriasis is recognized as the onset and exacerbation of psoriasis which is associated with certain drugs. It is often difficult to identify drug-related causes of psoriasis in clinical situations. This is because the latency period between the start of the medication and the onset of psoriatic skin lesions can vary considerably between drugs [32]. In some cases, the psoriasis flare can persist even after the suspected drug has been discontinued. Moreover, there may be little difference between psoriasis and drug-related psoriasis in terms of the clinical and histopathological findings [32]. Drug-related psoriasis would manifest as plaque psoriasis, palmoplantar psoriasis, nail psoriasis, scalp psoriasis, pustular psoriasis, and erythrodermic psoriasis [33]. In most cases, histopathological findings of drug-related psoriasis are virtually indistinguishable from those of conventional psoriasis [32]. Histopathological findings of eosinophilic infiltrates in the dermis and lichenoid reaction might help in the diagnosis of drug-related psoriasis [34]. While in psoriasis plaques unrelated to drugs, the capillaries in the upper dermis are convoluted and tortuous, that alteration is sometimes missing in drug-related psoriasis [34]. Moreover, there might also be differences regarding the formation of micro-abscesses of neutrophils in the upper layer of the epidermis [34]. However, these are just a few and not the most important clues that might orientate to a drug-related cause of psoriasis. Drug ingestion may result in the exacerbation of pre-existing psoriasis, induction of psoriatic lesions on clinically uninvolved skin in patients with psoriasis, or precipitation of the disease in patients without family history of psoriasis as well as in predisposed individuals [35]. The most widely accepted drugs are β-blockers, lithium, anti-malarial drugs, interferons, imiquimod, angiotensin-converting enzyme inhibitors, terbinafine, tetracycline, nonsteroidal anti-inflammatory drugs, and fibrate drugs [32,33,36,37]. The mechanisms of drug-related psoriasis still remain to be fully elucidated and the molecular mechanisms are complicated. However, some drugs have been known to affect keratinocyte hyperproliferation and the IL-23/IL-17 axis. Cyclic adenosine monophosphate (cAMP) is an intracellular messenger that is responsible for the stimulation of proteins for cellular differentiation and inhibition of proliferation, and β-blockers lead to a decrease in intraepidermal cAMP, causing keratinocyte hyperproliferation [32,33,38]. Imiquimod-induced skin inflammation is the most widely accepted psoriasis animal model [39]. Imiquimod, which activates the toll-like receptor-7/8, can induce and exacerbate psoriasis, critically dependent on the IL-23/IL-17 axis [39]. Recently, immune check point inhibitors and molecular inhibitors have been used for the treatment of malignancies and autoimmune diseases, and these drugs may affect the immune system, leading to the development of psoriasis [40,41,42]. The symptoms of psoriasis are rarely exacerbated during biologic therapy. However, psoriasis can also be triggered by biologics [43,44], and this is recognized as paradoxical reactions. Although most of the paradoxical reactions reported have been associated with the use of tumor necrosis factor (TNF)-α inhibitors, other biologics targeting interleukin (IL)-23 and IL-17 are increasingly common [45]. Biologics targeting TNF-α, IL-23, and IL-17 block immune signaling pathways, which can lead to cytokine imbalances [45]. Paradoxical reactions are thought to be due to an imbalance in cytokine production with an overproduction of IFN-α and altered lymphocyte recruitment and migration [45,46]. Suspected drugs should be discontinued and switched to an alternative drug in patients with drug-related psoriasis.

### 2.4. Vaccination

Patients with psoriasis are at increased risk of infection, mostly because of treatment with immunomodulatory or immunosuppressive drugs [47]. Thus, vaccination is recommended to prevent specific infections [47,48,49]. However, vaccination can often trigger and exacerbate psoriasis. Several studies support the association between influenza vaccination and the exacerbation of psoriasis [50,51]. Influenza vaccination may also trigger the onset of psoriasis [52]. Bacillus Calmette–Guerin (BCG) vaccine, which is a live attenuated strain of *Mycobacterium bovis*, is primarily used for the prevention of tuberculosis [53]. Psoriasis can be triggered post BCG vaccination [54,55]. BCG has also been used as local immunotherapy for bladder cancer, and a case of erythrodermic pustular psoriasis induced by BCG immunotherapy has been reported [56]. In a retrospective study, psoriasis was found to more frequently occur after adenovirus vaccination [57]. Psoriasis may also be triggered by other vaccines such as tetanus–diphtheria vaccination and pneumococcal polysaccharide vaccination [58,59]. These vaccinations are thought to generate T helper 1 (Th1) and Th17 immune responses which lead to the onset and exacerbation of psoriasis, although the precise pathomechanisms of psoriasis induced by vaccination remain to be elucidated. The incidence of psoriasis induced by vaccination is very low; rather, vaccination is therapeutically effective in patients with psoriasis.

### 2.5. Infection

The association between psoriasis and streptococcal infection is well established [60]. Psoriasis occurs after streptococcal infection, and the most common type is guttate psoriasis. Although the symptoms are self-limited, they can recur with the recurrence of streptococcal infection. Thus, tonsillectomy may be a potential treatment option for patients with recalcitrant psoriasis associated with episodes of tonsillitis [61]. Although prior infection with *Streptococcus pyogenes* is associated with guttate psoriasis, the ability to trigger guttate psoriasis is not serotype specific [60]. *Staphylococcus (S.) aureus* is also associated with the development of psoriasis [38]. Dysregulated skin microbiomes have been found to be associated with psoriasis [62]. Colonization of *S. aureus* in the lesions has been demonstrated in approximately 60% of patients with psoriasis, compared with 5% to 30% of normal healthy skin [38]. Moreover, the severity of psoriasis significantly correlates with enterotoxin production by the isolated *S. aureus* strains [63]. *Candida* species are a part of the normal human microbiota, and they were highly detected in either the skin or the mucosal membranes of patients with psoriasis [64]. A statistically significantly higher *Candida* species detection rate was also observed for mucosal membranes [64]. The detection rates of *Candida* species are significantly higher in patients with psoriasis as compared with those in healthy controls, especially in the oral mucosa milieux [64]. However, patients with psoriasis and healthy controls do not significantly differ in the rate of *Candida* species isolated from the skin [64]. *Candida albicans* is the most common disease-causing *Candida* species and its colonization promotes antifungal immunity, which may be associated with the pathogenesis of psoriasis [65]. *Malassezia* is a lipophilic yeast found on skin and body surfaces; it may contribute to the exacerbation of psoriasis [38]. It still remains to be established whether the species of *Malassezia* can initiate the development of psoriasis lesions. Human immunodeficiency virus (HIV) is also a well-known risk factor associated with psoriasis [66]. It is paradoxical that, while drugs that target T lymphocytes are effective in psoriasis, the condition should be exacerbated by HIV infection [66]. Although HIV infection causes the onset and exacerbation of psoriasis, the precise pathomechanisms still remain to be fully elucidated. Other viruses such as papilloma viruses, retroviruses, and endogenous retroviruses have also been implicated in psoriasis [67].

### 2.6. Lifestyle

Smoking and alcohol consumption have been associated with psoriasis. A systematic review and meta-analysis revealed that patients with psoriasis are more likely to be current or former smokers [68]. Smoking is associated with an increased risk of developing psoriasis [69]. In addition, smoking is strongly associated with pustular lesions of psoriasis [70]. A trend was found toward an increased risk of psoriasis with increasing pack-years or duration of smoking. Another study also showed that there was a positive correlation between the amount and/or duration of smoking and the occurrence of psoriasis [71]. Alcohol consumption appears to be a risk factor for psoriasis. However, a past systematic review concluded that there was not enough evidence to establish whether the alcohol consumption was indeed a risk factor [72]. Nonetheless, alcohol consumption was observed to be greater in patients with psoriasis than in the general population. Although the relationship between psoriasis and alcohol consumption is complex and multifactorial, alcohol abuse positively correlates with psoriasis severity and reduced treatment efficacy [73]. In addition, alcohol abuse is associated with significantly increased mortality rates [74]. Qualitative changes to the diet may play a significant role in maintaining the intestinal microbiome, and diet-induced dysbiosis may induce the cytokine imbalances associated with the pathogenesis of psoriasis [75,76,77]. Dietary modifications such as supplementation with polyunsaturated fatty acids, folic acid, vitamin D, and antioxidants can also be considered as adjuncts in the management of psoriasis [73]. To date, randomized controlled trials have produced conflicting results. Diet is a complex combination of foods from various groups; nutrients and the rich diversity of such foods may contribute to its protective effects against psoriasis [78].

## 3. Intrinsic Risk Factors

### 3.1. Obesity

Metabolic syndrome is common in patients with psoriasis [79,80,81,82,83] and obesity is strongly associated with the onset and exacerbation of psoriasis [78,84,85]. Patients with psoriasis have a significantly higher prevalence of obesity [70,86,87,88] as well as a higher risk of obesity [89,90,91]. In a previous meta-analysis, obesity was associated with severe psoriasis [92]. A large prospective cohort study also showed a positive association between body mass index (BMI) and psoriasis [93]. However, BMI has high specificity but low sensitivity to identify adiposity, as it fails to identify half of the people with excess body fat [94,95]. In contrast, waist circumference is more reliable measure of body fat, and many studies have shown a strong association between waist circumference and psoriasis [93,96,97]. Obesity can be defined as the expansion of white adipose tissue [85], and various mediators secreted by adipose tissue lead to a low-grade inflammatory state, contributing to the pathogenesis of psoriasis [98,99,100,101]. Pro-inflammatory adipokines such as TNF-α, IL-6, leptin, and adiponectin are produced in adipose tissue [98]. Blocking the TNF-α signaling pathway improves the inflammatory cycle of psoriasis, while it does not improve insulin sensitivity in patients with type 2 DM [102]. Leptin is an adipose tissue hormone that functions as an afferent signal in a negative feedback loop that maintains homeostatic control of adipose tissue mass [103]. Leptin is an important regulator of metabolic status and influences inflammatory and immune responses [104]. Leptin can enhance immune functions, including inflammatory cytokine production in macrophages, granulocyte chemotaxis, and increased Th17 proliferation [105,106]. The presence of elevated leptin inhibits the differentiation of regulatory T cells, which maintain tolerance and prevent psoriasis, in adipose tissue [106]. In fact, serum or plasma levels of leptin are higher in patients with psoriasis as compared to the healthy controls [107]. In addition, tissue levels of leptin are increased in the skin of patients with psoriasis [108]. Adiponectin is an adipocyte-specific factor which contributes to a beneficial metabolic action in whole-body energy homeostasis [109]. In contrast to leptin, adiponectin protects cells from apoptosis and reduces inflammation in various cell types [109]. Although adiponectin may act as an anti-inflammatory adipokine in patients with psoriasis, the association still remains unclear [110]. Weight loss itself appears to improve psoriasis symptoms [111,112] and is likely to improve decreased response to oral systemic therapies and biologics [113,114,115,116,117]. Moreover, weight loss may decrease the risk of drug toxicity of systemic therapies [118,119,120,121].

### 3.2. Diabetes Mellitus

The prevalence of DM is generally influenced by ethnic origin and lifestyle factors. However, the prevalence of DM might be similar among diverse patient populations, ethnic backgrounds, and baseline therapy [122]. A meta-analysis revealed that psoriasis was associated with DM [122]. Other meta-analyses have also demonstrated the association between psoriasis and the risk of DM [81,123]. DM is divided into two groups, namely, type 2 and type 1 DM. Patients with psoriasis have a significantly higher risk of type 2 DM. However, the prevalence of type 2 DM does not correlate with patient age or severity of psoriasis [124]. Psoriasis is a marker for increased risk of type 2 DM independent of its severity. It is unclear which disease comes first, psoriasis or type 2 DM [124]. As mentioned above, obesity is a risk factor for psoriasis. Obesity contributes to the onset and exacerbation of type 2 DM directly. Thus, obesity is associated with psoriasis as well as type 2 DM, and type 2 DM may not contribute to the pathogenesis of psoriasis directly. In contrast to type 2 DM, type 1 DM is a chronic disease characterized by insulin deficiency due to autoimmune destruction of insulin-producing pancreatic β-cells, leading to hyperglycemia [125]. Proinflammatory cytokines, including TNF-α, are involved in the pathogenesis of type 1 DM [126,127]. Interestingly, both Th1 and Th17 cells may contribute to the onset of type 1 DM [128,129]. Although type 1 DM may not contribute to the pathogenesis of psoriasis directly, the TNF-α/IL-23/IL-17 axis plays a crucial role in the pathogenesis of psoriasis and type 1 DM.

### 3.3. Dyslipidemia

Psoriasis is associated with obesity [70,86,87,88], and excess adipose tissue may contribute to dyslipidemia. Patients with psoriasis have a higher prevalence of dyslipidemia, which is likely to increase with the severity of psoriasis [130,131,132,133]. A past study including 70 patients with psoriasis revealed that dyslipidemia was observed in 62.85% of the patients [133]. Most often it was hypertriglyceridemia (39%) and hypertriglyceridemia with a lowered value of high-density lipoprotein (HDL). Dyslipidemia can also appear during oral systemic therapies for psoriasis [134]. Retinoids have the most potent activity leading to dyslipidemia, such as increased levels of triglycerides, total cholesterol, low-density lipoprotein cholesterol, and very-low-density lipoprotein cholesterol and simultaneously decreased levels of HDL cholesterol [135,136,137]. Cyclosporin can also lead to dyslipidemia [138]. It is possible that cyclosporine unmasks a latent tendency for mild to moderate hypertriglyceridemia [138], and this study concluded that fasting triglyceride levels should be monitored during cyclosporine therapy, especially after 1 to 2 months of use, and in patients with preexisting increased triglycerides and/or a history of etretinate use. Although dyslipidemia is associated with immunological abnormalities [134], it still remains unknown whether dyslipidemia affects the onset and exacerbation of psoriasis.

### 3.4. Hypertension

In a meta-analysis, patients with psoriasis showed greater prevalence and incidence of hypertension [139]. This meta-analysis also revealed that severe psoriasis was associated with greater incidence of hypertension [139]. Patients with psoriasis appear to have more severe hypertension [140,141]. A multicenter noninterventional observational study including 2210 patients with psoriasis revealed that 26% of patients with psoriasis had hypertension, and the incidence of hypertension was higher when compared with the general population [142]. Conversely, hypertension may be associated with the incidence of psoriasis [143]. Although psoriasis and hypertension have shared risk factors such as obesity and smoking, most studies have shown an independent association of psoriasis with hypertension after adjusting for these risk factors [139]. The mechanisms underlying this association remain unknown.

### 3.5. Mental Stress

Mental stress is a feeling of strain and pressure caused by internal perceptions which lead to anxiety or other negative emotions. Mental stress occurs when individuals think the demands exceed their ability to cope. Mental stress is commonly regarded as a well-established trigger of psoriasis and many patients with psoriasis and physicians believe that mental stress exacerbates psoriasis. Although psoriasis leads to higher degree of distress as proved by measurements on Dermatology Life Quality Index scales, the relation between mental stress and psoriasis is complex. In a past systematic review including 39 studies (32,537 patients), 46% of patients believed their disease was stress reactive and 54% recalled preceding stressful events [144]. However, there was no high-quality evidence to support the notion that the preceding stress was strongly associated with the onset and exacerbation of psoriasis. The association was based primarily on retrospective studies with many limitations. It seems unclear whether mental stress affects the clinical course of psoriasis. In contrast, a prospective study concluded that cognitive and behavioral patterns of worrying and scratching were both independently related to an increase four weeks later in disease severity and itch, at moments when patients experienced a high level of daily stressors [145]. At these moments, stressors also interacted with vulnerability factors, suggesting that patients with more daily stress and high worrying and scratching had particularly worsened disease severity and itch. Scratching in response to itch subsequently leads to an itch–scratch–itch cycle causing the exacerbation of psoriasis. Further studies are necessary to elucidate the association between mental stress and psoriasis.

## 4. Conclusions

In this review, both extrinsic and intrinsic risk factors for the development of psoriasis were discussed in detail. Biologics have dramatically changed the treatment of psoriasis. In contrast, elimination of the risk factors is also important for controlling the disease. From the clinicians’ perspective, the exacerbation of psoriasis induced by the Koebner phenomenon and drugs can be avoided by proper knowledge. From the patients’ perspective, lifestyle could be modified by proper education, although the extent varies among patients. However, various factors interact with each other and can affect the pathogenesis of psoriasis directly and/or indirectly. For example, obesity, dyslipidemia, and hypertension are associated with the course of psoriasis and are also dependent on the patient’s age, lifestyle, and concomitant diseases. Moreover, the impacts of the patient’s age, lifestyle, and concomitant diseases vary among individuals. The risk factors of psoriasis are not fully understood, and future studies need to successfully establish preventive approaches for psoriasis.

## Figures and Tables

**Figure 1 ijms-20-04347-f001:**
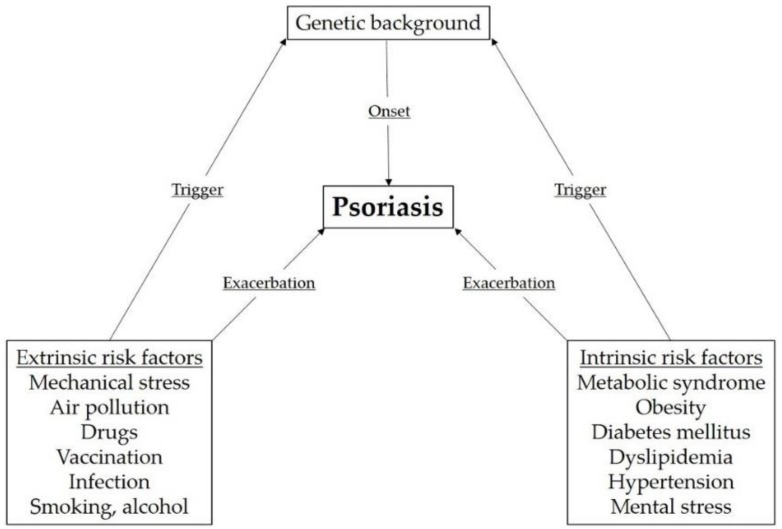
Risk factors for the onset and exacerbation of psoriasis. As shown in this figure, extrinsic and intrinsic factors are associated with the onset and exacerbation of psoriasis.

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
