# Peer review of "Risk Factors for the Development of Psoriasis"

_ijms, 2019, doi:10.3390/ijms20184347_

Round 1
Reviewer 1 Report
Well documented paper summarizing extrinsic and intrinsic risk factors for the development of psoriasis.Author Response
Comments and Suggestions for Authors:
Well documented paper summarizing extrinsic and intrinsic risk factors for the development of psoriasis.
Response: Thank you very much for favorably reviewing our manuscript.
Reviewer 2 Report
The article aims to present the factors involved in psoriasis onset and/or flares, grouping all factors regarding their relation with the internal/external milieu, namely intrinsic/extrinsic factors. Although the presented factors are well established and undoubtedly familiar to every dermatology practitioner, the article brings somehow an updated and refreshed perspective, the information is presented in a structured manner, and the work is thoroughly documented. Molecular interplays are quite well described, in an accesible manner.
The following changes are suggested to the authors:
Line98—please avoid repetition in “When drug-induced psoriasis is suspected, suspected drugs.....”
Line 80 “However, histopathological findings of eosinophilic infiltrates in the dermis and lichenoid reaction may help in the diagnosis of drug-induced psoriasis” – authors should also take into account that these are just a few–and not the most important- clues that might orientate the dermatologist to a drug-related cause of psoriasis. While in psoriasis plaques unrelated to drugs the cappilaries in upper dermis are convoluted and tortuous, that alteration is sometimes missing in drug-related psoriasis; more subtle changes may affect suprapapillary epidermis; moreover, there might also be differeces regarding the formation of micro-abscesses of neutrophils in the upper layer of the epidermis; please include these microscopic changes as well, unbiased, as they are reflected in the current literature on this particular subject.
Line 86 “ Fibrate drugs may also affect the development of psoriasis [31, 32].”-please include fibrates in the anterior sentence, within the list of drugs, as the effect of fibrates on psoriasis does not exceed that of the aforementioned drugs; therefore it would be rather fair fibrates not deserve a separate place in the paragrph, apart of the list in the previous sentence.
Line24 “ The prevalence in adults 24 ranges from 0.51 to 11.43 % while that in children ranges from 0 to 1.37 % [4]” –might be rephrased
Line 26 “These observations suggest that ethnicity, genetic background, and environmental factors affect the onset of psoriasis” –please be more specific about “these observations” .
“drug”--> should be changed to “drugs”, as there is a multitude of drugs that can elicit psoriasis onset/ aggravation; this refers to –but not exclusively- the title of paragraph 2.2 and to Figure 1, left box.
Line 60 “Resident memory T cells (TRM) are recently described subset” --> please check grammar
Line 77 “Drug-induced psoriasis is observed as plaque psoriasis, palmoplantar psoriasis” --> should be rephrased as psoriasis would manifest as[...], rather than just be observed as[...]
Line 101- a short documented argumentation in a few phrases to support the sentence “Patients with psoriasis are at increased risk of infection, partially because of the disease itself” would be advised, as it would be beneficial to the reader: please state why in an argumented manner.
Paragraph 2.5 – please specify that when refered to diet, the authors have in mind, in this paragraph, only qualitative changes; quantitative changes are not taken into consideration in this particular paragraph (quantitative changes in alimentation are of paramount importance in psoriasis, especially on long term – e.g., hypercaloric diet leads to obesity, the latter impacting psoriasis in various ways etc) and that should be stated clearly, in order to avade any possible confusion.
A paragraph/short specification should be added regarding the possibility to modify each factor; not all factors could be modified by individuals receiving proper education, however some of them could be modified (e.g by lifestyle changes); which factors are modifiable, and to which extent? adding this is particularly important, as this may add benefit to the clinical dermatologist reading the article.
Line 261 “... the age of onset of psoriasis was significantly higher in the hypertension group when compared with general population” --> Indeed this is the finding in the cited French study; Hoever, this part is quite intriguing, as it may be interpreted like arterial hypertension acts rather as a protective factor/delaying factor, while having hypertension leads to a delay in the onset of psoriasis. This sentence would need further expalining/argumentation..
Paragraph 3.5 – Authors discuss over the influence of percieved stress on psoriasis. It worth mentioning that the relation between mental stress and psoriasis is complex, as psoriasis leads to higher degree of distress, as proved by measurements on DLQI scales.
However, the authors do not state anything about the effect of UV radiation (natural/artificial sources)
While they cite sun exposure as a precipitating factor in Lines 33-34
“ the exacerbating factors for the Japanese population were observed to be stress (6.4 to 16.6%), seasonal factors (9.7 to 13.3%), [,....], sun exposure (1.3 to 3.5%), and B-blockers (0.9 to 2.3%)”, the article still contains no more explanations regarding sun effect on psoriasis whatsoever.
This is rather important, as even the authors themselves cite sun exposure and seasonal factors as main triggers for psoriasis, but only tangentially, without any further details or analyze.
Sun exposure deserves a separate paragraph,- or a sub-paragraph within a bigger paragraph to be entitled physical factors (that would include mechanical factors –Koebner reaction, etc and UV radiation)
UV role in psoriasis worsening may be paradoxical, as UV treatment is well established in the therapeutic arsenal for psoriasis for centuries; therefore, this apparently dual role of UV in triggering/healing psoriasis may worth being mentioned.
Moreover, the article would benefit from including a few data on so-called photosensitive psoriasis, a subset of the disease that seems to be predominent in female patiens, associated with HLA-Cw*/0602, exhibiting seasonal flares and aggravation after sun exposure.
Moreover, the manuscript would be improved if the authors would also include, beside UV radiation as triggering factor, environment pollution as extrinsic factor, at least in a few sentences. Current medical literature in the field contains a handful of studies that link psoriasis to cadmium pollution level, chromium, nickel, zinc from cement factories, Arsenic, and other pollutants and this should be discussed over a few sentences.
Overall, by summing up a large quantity of evidence-based studies, the article might be a useful contribution to the journal. I recommend the article to be published, after all suggested changes are taken into consideration by authors.
Author Response
Comments and Suggestions for Authors:
The article aims to present the factors involved in psoriasis onset and/or flares, grouping all factors regarding their relation with the internal/external milieu, namely intrinsic/extrinsic factors. Although the presented factors are well established and undoubtedly familiar to every dermatology practitioner, the article brings somehow an updated and refreshed perspective, the information is presented in a structured manner, and the work is thoroughly documented. Molecular interplays are quite well described, in an accesible manner.
Response: We would like to express our sincere thanks for your efforts on this matter in advance. We are grateful for the detailed evaluation. We are also grateful for the contributions obtained from the reviewers and the opportunity to improve the quality of our manuscript.
The following changes are suggested to the authors:
Point 1: Line98—please avoid repetition in “When drug-induced psoriasis is suspected, suspected drugs.....”
Response 1: We thank the reviewer for pointing this issue. In the new manuscript, we have revised our manuscript as follows; Suspected drugs should be discontinued and switched to an alternative drug in patients with drug-related psoriasis (the paragraph “2.3. Drugs”).
Point 2: Line 80 “However, histopathological findings of eosinophilic infiltrates in the dermis and lichenoid reaction may help in the diagnosis of drug-induced psoriasis” – authors should also take into account that these are just a few–and not the most important- clues that might orientate the dermatologist to a drug-related cause of psoriasis. While in psoriasis plaques unrelated to drugs the cappilaries in upper dermis are convoluted and tortuous, that alteration is sometimes missing in drug-related psoriasis; more subtle changes may affect suprapapillary epidermis; moreover, there might also be differeces regarding the formation of micro-abscesses of neutrophils in the upper layer of the epidermis; please include these microscopic changes as well, unbiased, as they are reflected in the current literature on this particular subject.
Response 2: We thank the reviewer for pointing this issue and your suggestion. In the new manuscript, we have incorporated the following sentences as the reviewer suggests; Histopathological findings of eosinophilic infiltrates in the dermis and lichenoid reaction might help in the diagnosis of drug-related psoriasis. While in psoriasis plaques unrelated to drugs the capillaries in upper dermis are convoluted and tortuous, that alteration is sometimes missing in drug-related psoriasis. Moreover, there might also be differences regarding the formation of micro-abscesses of neutrophils in the upper layer of the epidermis. However, these are just a few and not the most important clues that might orientate to a drug-related cause of psoriasis (the paragraph “2.3. Drugs”).
Point 3: Line 86 “ Fibrate drugs may also affect the development of psoriasis [31, 32].”-please include fibrates in the anterior sentence, within the list of drugs, as the effect of fibrates on psoriasis does not exceed that of the aforementioned drugs; therefore it would be rather fair fibrates not deserve a separate place in the paragrph, apart of the list in the previous sentence.
Response 3: We thank the reviewer for pointing this issue. In the new manuscript, we have revised our manuscript as the reviewer suggests (the paragraph “2.3. Drugs”).
Point 4: Line24 “ The prevalence in adults 24 ranges from 0.51 to 11.43 % while that in children ranges from 0 to 1.37 % [4]” –might be rephrased
Response 4: We thank the reviewer for pointing this issue and your suggestion. In the new manuscript, we have deleted this sentence to address the point 5 comment (the paragraph “1. Introduction”).
Point 5: Line 26 “These observations suggest that ethnicity, genetic background, and environmental factors affect the onset of psoriasis” –please be more specific about “these observations” .
Response 5: We thank the reviewer for pointing this issue and your suggestion. In the new manuscript, we have revised our manuscript as follows; The prevalence of psoriasis varies with the country and psoriasis can appear at any age, suggesting that ethnicity, genetic background, and environmental factors affect the onset of psoriasis (the paragraph “1. Introduction”).
Point 6: “drug”--> should be changed to “drugs”, as there is a multitude of drugs that can elicit psoriasis onset/ aggravation; this refers to –but not exclusively- the title of paragraph 2.2 and to Figure 1, left box.
Response 6: We thank the reviewer for pointing this issue. In the new manuscript, we have revised our manuscript as the reviewer suggests.
Point 7: Line 60 “Resident memory T cells (TRM) are recently described subset” --> please check grammar
Response 7: We thank the reviewer for pointing this issue. In the new manuscript, we have revised our manuscript as follows; Resident memory T cells (TRM) have been described as a non-circulating memory T cell subset that persists long-term in peripheral tissues (the paragraph “2.1. Mechanical stress”).
Point 8: Line 77 “Drug-induced psoriasis is observed as plaque psoriasis, palmoplantar psoriasis” --> should be rephrased as psoriasis would manifest as[...], rather than just be observed as[...]
Response 8: We thank the reviewer for pointing this issue. In the new manuscript, we have revised our manuscript as the reviewer suggests (the paragraph “2.3. Drugs”).
Point 9: Line 101- a short documented argumentation in a few phrases to support the sentence “Patients with psoriasis are at increased risk of infection, partially because of the disease itself” would be advised, as it would be beneficial to the reader: please state why in an argumented manner.
Response 9: We thank the reviewer for pointing this issue. This sentence seemed confusing to the readers and we have revised our manuscript as follows; Patients with psoriasis are at increased risk of infection, mostly because of the treatment with immunomodulatory or immunosuppressive drugs (the paragraph “2.4. Vaccination”).
Point 10: Paragraph 2.5 – please specify that when refered to diet, the authors have in mind, in this paragraph, only qualitative changes; quantitative changes are not taken into consideration in this particular paragraph (quantitative changes in alimentation are of paramount importance in psoriasis, especially on long term – e.g., hypercaloric diet leads to obesity, the latter impacting psoriasis in various ways etc) and that should be stated clearly, in order to avade any possible confusion.
Response 10: We thank the reviewer for pointing this issue. In the new manuscript, we have used the term ”qualitative changes of diet” as the reviewer suggests (the paragraph “2.6. Lifestyle”).
Point 11: A paragraph/short specification should be added regarding the possibility to modify each factor; not all factors could be modified by individuals receiving proper education, however some of them could be modified (e.g by lifestyle changes); which factors are modifiable, and to which extent? adding this is particularly important, as this may add benefit to the clinical dermatologist reading the article.
Response 11: We thank the reviewer for pointing this issue and your suggestion. In the conclusion paragraph, we have incorporated the following sentences; From the clinicians’ perspective, the exacerbation of psoriasis induced by Koebner phenomenon and drugs could be avoided by proper knowledge. From the patients’ perspective, lifestyle changes could be modified by proper education, although the extent varies among patients (the paragraph “4. Conclusions”).
Point 12: Line 261 “... the age of onset of psoriasis was significantly higher in the hypertension group when compared with general population” --> Indeed this is the finding in the cited French study; Hoever, this part is quite intriguing, as it may be interpreted like arterial hypertension acts rather as a protective factor/delaying factor, while having hypertension leads to a delay in the onset of psoriasis. This sentence would need further expalining/argumentation.
Response 12: We thank the reviewer for pointing this issue. We have to apologize in regard to this point. From the findings of this study, the age of onset was significantly higher in the hypertension group, when compared with familial psoriasis. In the new manuscript, we have revised our manuscript as follows; A multicenter noninterventional observational study including 2,210 patients with psoriasis revealed that 26% of patients with psoriasis had hypertension and the incidence of hypertension was higher when compared with general population (the paragraph “3.4. Hypertension”).
Point 13: Paragraph 3.5 – Authors discuss over the influence of percieved stress on psoriasis. It worth mentioning that the relation between mental stress and psoriasis is complex, as psoriasis leads to higher degree of distress, as proved by measurements on DLQI scales.
Response 13: We thank the reviewer for pointing this issue. In the new manuscript, we have incorporated the following sentence as the reviewer suggests; Although psoriasis leads to higher degree of distress as proved by measurements on Dermatology Life Quality Index scales, the relation between mental stress and psoriasis is complex (the paragraph “3.5. Mental stress”).
Point 14: However, the authors do not state anything about the effect of UV radiation (natural/artificial sources)
While they cite sun exposure as a precipitating factor in Lines 33-34
“ the exacerbating factors for the Japanese population were observed to be stress (6.4 to 16.6%), seasonal factors (9.7 to 13.3%), [,....], sun exposure (1.3 to 3.5%), and B-blockers (0.9 to 2.3%)”, the article still contains no more explanations regarding sun effect on psoriasis whatsoever.
This is rather important, as even the authors themselves cite sun exposure and seasonal factors as main triggers for psoriasis, but only tangentially, without any further details or analyze.
Sun exposure deserves a separate paragraph,- or a sub-paragraph within a bigger paragraph to be entitled physical factors (that would include mechanical factors –Koebner reaction, etc and UV radiation)
UV role in psoriasis worsening may be paradoxical, as UV treatment is well established in the therapeutic arsenal for psoriasis for centuries; therefore, this apparently dual role of UV in triggering/healing psoriasis may worth being mentioned.
Moreover, the article would benefit from including a few data on so-called photosensitive psoriasis, a subset of the disease that seems to be predominent in female patiens, associated with HLA-Cw*/0602, exhibiting seasonal flares and aggravation after sun exposure.
Response 14: We thank the reviewer for pointing this issue and your suggestion. In the new manuscript, we have added a new paragraph (2.2. Air pollutants and sun exposure) and discussed about the relation between air pollution (air pollutants and sun exposure) and psoriasis. In addition, we have added the description about sun exposure in the pathogenesis of psoriasis as the reviewer suggests.
Point 15: Moreover, the manuscript would be improved if the authors would also include, beside UV radiation as triggering factor, environment pollution as extrinsic factor, at least in a few sentences. Current medical literature in the field contains a handful of studies that link psoriasis to cadmium pollution level, chromium, nickel, zinc from cement factories, Arsenic, and other pollutants and this should be discussed over a few sentences.
Overall, by summing up a large quantity of evidence-based studies, the article might be a useful contribution to the journal. I recommend the article to be published, after all suggested changes are taken into consideration by authors.
Response 15: We thank the reviewer for pointing this issue and your suggestion. In the new manuscript, we have added a new paragraph (2.2. Air pollutants and sun exposure) and a few sentences about the relation between air pollutants and psoriasis.

Reviewer 3 Report
This review manuscript by Kamiya et al describes extrinsic and intrinsic risk factors for psoriasis development. The authors emphasize extrinsic factors including mechanical stress, drug, vaccination, infection and life style, as well as intrinsic factors including obesity, diabetes, dyslipidemia, hypertension and mental stress. Overall this manuscript is well written and highlight the triggering factors that initiate psoriasis and their roles in promoting activation of the TNF/IL23/IL17 axis in psoriasis pathogenesis. However, there are still a few concerns and please see specific comments below:
One major concern is that the authors should include more experimental data explaining molecular mechanisms underlying psoriasis pathogenesis by these risk factors. In this current version, mechanism has been explained well only for how “Mechanical stress” in section 2.1. But how psoriasis can be triggered by drugs (such as beta-blocker), infection, smoking, alcohol consumption or mental stress has not been explained. In section 2.1., the authors only describe the role of NGF (nerve growth factor) in psoriasis development after a cutaneous trauma (Koebner phenomenon), but other factors, such as IFNB and IFNA, have been reported to be associated with Koebner phenomenon. Smoking is an important triggering factor for psoriasis, but the author uses “lifestyle” for both smoking and alcohol. These two factors should be included and listed in lifestype in Figure 1 to highlight the importance of smoking in psoriasis pathogenesis. The section for “Diabetes mellitus” is quite confusing. The authors should explain better about how type 1 and/or type 2 diabetes may trigger psoriasis.Author Response
Comments and Suggestions for Authors:
This review manuscript by Kamiya et al describes extrinsic and intrinsic risk factors for psoriasis development. The authors emphasize extrinsic factors including mechanical stress, drug, vaccination, infection and life style, as well as intrinsic factors including obesity, diabetes, dyslipidemia, hypertension and mental stress. Overall this manuscript is well written and highlight the triggering factors that initiate psoriasis and their roles in promoting activation of the TNF/IL23/IL17 axis in psoriasis pathogenesis. However, there are still a few concerns and please see specific comments below:
Response: We would like to express our sincere thanks for your efforts on this matter in advance. We are grateful for the detailed evaluation. We are also grateful for the contributions obtained from the reviewers and the opportunity to improve the quality of our manuscript.
Point 1: One major concern is that the authors should include more experimental data explaining molecular mechanisms underlying psoriasis pathogenesis by these risk factors. In this current version, mechanism has been explained well only for how “Mechanical stress” in section 2.1. But how psoriasis can be triggered by drugs (such as beta-blocker), infection, smoking, alcohol consumption or mental stress has not been explained.
Response 1: We thank the reviewer for pointing this issue and your suggestion. As the reviewer pointed out, we should include more experimental data explaining molecular mechanisms underlying psoriasis pathogenesis by these risk factors. However, the molecular mechanisms are complicated and have not been fully studied. Some factors were not significantly associated with the development of psoriasis in some past meta-analyses, although those are well established and undoubtedly familiar to clinicians. Thus, we pick up topics which have drawn attention about the risk factors for psoriasis and discuss them with recent published works especially from clinical aspects. In contrast, b-blockers and imiquimod have been known to affect keratinocyte hyperproliferation and the IL-23/IL-17 axis, respectively. In the new manuscript, we have incorporated the following sentences; The mechanisms of drug-induced psoriasis still remain to be fully elucidated and the molecular mechanisms are complicated. However, some drugs have been known to affect keratinocyte hyperproliferation and the IL-23/IL-17 axis. Cyclic adenosine monophosphate (cAMP) is an intracellular messenger that is responsible for the stimulation of proteins for cellular differentiation and inhibition of proliferation, and b-blockers lead to a decrease in intraepidermal cAMP causing keratinocyte hyperproliferation. Imiquimod-induced skin inflammation is the most widely accepted psoriasis animal model. Imiquimod, which activates the toll-like receptor-7/8, can induce and exacerbate psoriasis, critically dependent on the IL-23/IL-17 axis (the paragraph “2.3. Drugs”).
Point 2: In section 2.1., the authors only describe the role of NGF (nerve growth factor) in psoriasis development after a cutaneous trauma (Koebner phenomenon), but other factors, such as IFNB and IFNA, have been reported to be associated with Koebner phenomenon.
Response 2: We thank the reviewer for pointing this issue. In the new manuscript, we have incorporated the following sentences as regards IFN-b and IFN-a; Type 1 interferons (IFNs), such as IFN-a and IFN-b, have been suggested to play an indispensable role in initiating psoriasis during skin injury. Skin injury rapidly induces IFN-b from keratinocytes and IFN-a from dermal plasmacytoid dendritic cells through distinct mechanisms. Host antimicrobial peptide LL37 potentiates double-stranded RNA immune pathways and single-stranded RNA or DNA pathways in plasmacytoid dendritic cells. Production of type 1 IFNs induced by skin injury may explain the Koebner phenomenon (the paragraph “2.1. Mechanical stress”).
Point 3: Smoking is an important triggering factor for psoriasis, but the author uses “lifestyle” for both smoking and alcohol. These two factors should be included and listed in lifestype in Figure 1 to highlight the importance of smoking in psoriasis pathogenesis.
Response 3: We thank the reviewer for pointing this issue. In the new manuscript, we have modified Figure 1 as the reviewer suggests.
Point 4: The section for “Diabetes mellitus” is quite confusing. The authors should explain better about how type 1 and/or type 2 diabetes may trigger psoriasis.
Response 4: We thank the reviewer for pointing this issue. The section for “Diabetes mellitus” was quite confusing. In the new manuscript, we clearly describe the term ”type 1 DM” and the term “type 2 DM”. However, it is possible that some past meta-analyses included both patients with type 1 DM and patients with type 2 DM. Thus, in this situation, we also use the term “DM” in the new manuscript (the paragraph “3.2. Diabetes mellitus”).

Reviewer 4 Report
The etiopathogenesis of psoriasis has not been fully investigated yet. The condition has been classified as an immune-mediated disease. Thus, the search for additional factors that trigger its development currently constitutes a significant clinical problem. Moreover, identification of risk factors both for the development of psoriasis and its exacerbation is synonymous to identifying factors that decide about patients’ quality of life and the effectiveness of treatment.
The paper presents an untypical structure. The Authors have divided psoriasis risk factors into external and internal. However, they did not attempt to thoroughly determine relationships between these factors. They could have done it based on, for example, the diagram in figure 1.
Furthermore, the paper lack clear presentation of risk factors (e.g. in tables) in the context of age, sex, selected medications.
The Conclusions do not correspond to the content of the paper. Given the current state of knowledge, it is controversial to state that prevention of psoriasis will be possible via elimination of its risk factors.
In my opinion, the paper submitted for review does not contribute anything new in terms of the information itself and the way it has been presented.
I do not recommend this paper for publication.
Author Response
Comments and Suggestions for Authors:
The etiopathogenesis of psoriasis has not been fully investigated yet. The condition has been classified as an immune-mediated disease. Thus, the search for additional factors that trigger its development currently constitutes a significant clinical problem. Moreover, identification of risk factors both for the development of psoriasis and its exacerbation is synonymous to identifying factors that decide about patients’ quality of life and the effectiveness of treatment.
Point 1: The paper presents an untypical structure. The Authors have divided psoriasis risk factors into external and internal. However, they did not attempt to thoroughly determine relationships between these factors. They could have done it based on, for example, the diagram in figure 1.
Response 1: We thank the reviewer for pointing this issue. We divided the risk factors for psoriasis into two groups, namely extrinsic and intrinsic risk factors. In this article, we did not discuss about the relationships between these components. For example, metabolic syndrome is a combination of diabetes, hypertension, and obesity. These components interact with each other, and can affect the pathogenesis of psoriasis directly and/or indirectly. Thus, we focused on each component of these groups.
Point 2: Furthermore, the paper lack clear presentation of risk factors (e.g. in tables) in the context of age, sex, selected medications.
Response 2: We thank the reviewer for pointing this issue. Based on the past studies, we could not determine the risk factors in the context of age, sex, and selected medication. Various factors affect the development of psoriasis and psoriasis can occur regardless of age or sex. Moreover, some factors were not significantly associated with the development of psoriasis in some past meta-analyses, although those are well established and undoubtedly familiar to clinicians.
Point 3: The Conclusions do not correspond to the content of the paper. Given the current state of knowledge, it is controversial to state that prevention of psoriasis will be possible via elimination of its risk factors.
In my opinion, the paper submitted for review does not contribute anything new in terms of the information itself and the way it has been presented.
I do not recommend this paper for publication.
Response 3: We thank the reviewer for pointing this issue. In this review article, we pick up topics which have drawn attention about the risk factors for psoriasis and discuss them with recent published works especially from clinical aspects. This article was evaluated strictly and we partly agree with your comments. But, we believe this article brings somehow an updated and refreshed perspective and will be of interest to the readers of the IJMS.
Reviewer 5 Report
ijms-565179
“Risk factors for the development of psoriasis” by Kamiya et al.
In this review the authors stated the extrinsic and intrinsic risk factors of psoriasis. The authors summarized the risk factors from several meta-analysis and clinical study. However, as the review of this journal, the authors should add molecular mechanisms based on their research.
Major points:
In this review, the authors referred only one study which was done by themselves. The review should be based on the authors research. Please add your studies. Sevela drugs induces drug-related psoriasis. The moclecular mechanisms should be discussed. line 87-88: The authors stated “rapid withdrawal of systemic corticosteroids frequently causes the exacerbation of psoriaisis.” It is true. But it was caused by enhancement of immune responses as rebound responses induced by decease in corticosteroids. I guess it was apparently different from drug-induced psoriasis. If possible, the authors should discuss the risk factors which causes onset (trigger) and those which causes exacerbation, separately. Is there any specific risk factors for psoriasis as compared with the other immune diseases and skin diseases?Author Response
Comments and Suggestions for Authors:
In this review the authors stated the extrinsic and intrinsic risk factors of psoriasis. The authors summarized the risk factors from several meta-analysis and clinical study. However, as the review of this journal, the authors should add molecular mechanisms based on their research.
Response: We would like to express our sincere thanks for your efforts on this matter in advance. We are grateful for the detailed evaluation. We are also grateful for the contributions obtained from the reviewers and the opportunity to improve the quality of our manuscript.
Major points:
Point 1: In this review, the authors referred only one study which was done by themselves. The review should be based on the authors research. Please add your studies.
Response 1: We thank the reviewer for pointing this issue. As the reviewer pointed out, this review should be based on our research. We have conducted a nationwide search for psoriasis in Japan. We have also studied about the risk factors for psoriasis. However, unfortunately, we have not published a paper about them at this point. Thus, we pick up topics which have drawn attention about the risk factors for psoriasis and discuss them with recent published works from clinical aspects.
Point 2: Sevela drugs induces drug-related psoriasis. The moclecular mechanisms should be discussed.
Response 2: We thank the reviewer for pointing this issue and your suggestion. Many drugs are thought to be associated with the development of psoriasis. The most widely accepted drugs are b-blockers, lithium, anti-malarial drugs, interferons, imiquimod, angiotensin-converting enzyme inhibitors, terbinafine, tetracycline, and nonsteroidal anti-inflammatory drugs. The mechanisms of drug-related psoriasis still remain to be fully elucidated and the molecular mechanisms are complicated. However, b-blockers and imiquimod have been known to affect keratinocyte hyperproliferation and the IL-23/IL-17 axis, respectively. In the new manuscript, we have incorporated the following sentences; The mechanisms of drug-related psoriasis still remain to be fully elucidated and the molecular mechanisms are complicated. However, some drugs have been known to affect keratinocyte hyperproliferation and the IL-23/IL-17 axis. Cyclic adenosine monophosphate (cAMP) is an intracellular messenger that is responsible for the stimulation of proteins for cellular differentiation and inhibition of proliferation, and b-blockers lead to a decrease in intraepidermal cAMP causing keratinocyte hyperproliferation. Imiquimod-induced skin inflammation is the most widely accepted psoriasis animal model. Imiquimod, which activates the toll-like receptor-7/8, can induce and exacerbate psoriasis, critically dependent on the IL-23/IL-17 axis (the paragraph “2.3. Drugs”).
Point 3: line 87-88: The authors stated “rapid withdrawal of systemic corticosteroids frequently causes the exacerbation of psoriaisis.” It is true. But it was caused by enhancement of immune responses as rebound responses induced by decease in corticosteroids. I guess it was apparently different from drug-induced psoriasis.
Response 3: We thank the reviewer for pointing this issue. As the reviewer pointed out, the exacerbation of psoriasis induced by rapid withdrawal of systemic corticosteroids may not be recognized as drug-induced psoriasis. In the new manuscript, we have deleted the following sentence: In addition, rapid withdrawal of systemic corticosteroids frequently causes the exacerbation of psoriasis [33].
Point 4: If possible, the authors should discuss the risk factors which causes onset (trigger) and those which causes exacerbation, separately.
Response 4: We thank the reviewer for pointing this issue. As the reviewer pointed out, we first discussed the risk factors which cause onset and those which cause exacerbation, separately. However, some factors are associated with the onset and exacerbation of psoriasis and others are associated with the onset or exacerbation of psoriasis. Thus, we divided risk factors into two groups and focused on each component of these groups in this review article.
Point 5: Is there any specific risk factors for psoriasis as compared with the other immune diseases and skin diseases?
Response 5: We thank the reviewer for pointing this issue. In this review, we focused on Koebner phenomenon, drugs, vaccination, infection, lifestyle, metabolic syndrome, and mental stress. These factors are also associated with other immune diseases and skin diseases in other mechanisms. However, some drugs may be a specific risk factor for psoriasis. For example, imiquimod-induced skin inflammation is currently the most widely accepted psoriasis animal model, as mentioned above.
Round 2
Reviewer 3 Report
The authors have adequately addressed my questions and concerns, and I have no more questions.
Author Response
The authors have adequately addressed my questions and concerns, and I have no more questions.
Response: Thank you very much for favorably reviewing our manuscript.

Reviewer 4 Report
The Authors have introduced a number of corrections in the paper. However, on the whole, they have not followed my remarks.
I would like to thank the Authors for correcting the paper.
However, I still think it should include age-specific factors. And so e.g.:
in infants - the use of disposable diapers and lesions in the genital area, vaccinations, infectious
diseases;
in children – streptococcal infections, helminthoses , caries dentium , the use of plaster dressings ,etc;
in women – menstruation, pregnancy, delivery, medications, alcohols;
in men – medications, alcohol, occupational risk of injuries at physical work;
in the elderly – diminished immune response, psoriasis in the third age
factors common to all patients suggestions. ( eg Childhood psoriasis
Dorota Piekarska-Myślińska1, Aldona Pietrzak1, Wojciech Myśliński1, Daniel Pietrzak1, Magdalena Borysowicz2,
Mateusz Socha3, Dorota Krasowska Dermatol Rev 2017, 104, 363–376
DOI: https://doi.org/10.5114/dr.2017.69944, )
Having reviewed the intrinsic risk factors for psoriasis presented in the paper, one may get the impression that there are no relationships between them. I cannot agree with such an opinion or the Authors’ approach towards the problem analysed because obesity, lipid abnormalities and hypertension jointly occur in a number of diseases, including diabetes, CVD and psoriasis.
The approach proposed does not take into account the concomitance, “overlap” of risk factors. The relationships between the extrinsic and intrinsic risk factors cannot be ignored.
I sustain my initial remark that the paper lacks clear presentation of risk factors (e.g. in tables) in the context of age, sex and selected medications. Please note that obesity, lipid abnormalities and hypertension are associated with the course of psoriasis and are also dependent on the patients’ age, lifestyle and concomitant diseases. Separate analysis for individual factors in chronic diseases is not the right approach towards the problem presented.
Conclusions have been modified and better summarise the problem in question.
Additionally, please note that the attempt to identify risk factors needs to be accompanied with their estimation and determination of their OR or HR values. The paper lacks a concise and hierarchical presentation of the parameters analysed. Thus, it does not indicate which of the parameters - based on the data/literature review - may be involved in the etiopathogenesis of psoriasis.
Again, I believe that in terms of the information itself and the way it has been presented the paper does not contribute anything new for clinicians or researchers studying psoriasis.
I would like authors to reshape the body of the article .
Author Response
Comments and Suggestions for Authors
Point 1: The Authors have introduced a number of corrections in the paper. However, on the whole, they have not followed my remarks. I would like to thank the Authors for correcting the paper. However, I still think it should include age-specific factors. And so e.g.: in infants - the use of disposable diapers and lesions in the genital area, vaccinations, infectious diseases; in children – streptococcal infections, helminthoses, caries dentium, the use of plaster dressings, etc; in women – menstruation, pregnancy, delivery, medications, alcohols; in men – medications, alcohol, occupational risk of injuries at physical work; in the elderly – diminished immune response, psoriasis in the third age factors common to all patients suggestions.
( eg Childhood psoriasis Dorota Piekarska-Myślińska1, Aldona Pietrzak1, Wojciech Myśliński1, Daniel Pietrzak1, Magdalena Borysowicz2, Mateusz Socha3, Dorota Krasowska Dermatol Rev 2017, 104, 363–376 DOI: https://doi.org/10.5114/dr.2017.69944, )
Response 1: We thank the reviewer for pointing this issue and your suggestion. We can understand your comments from this explanation. Various factors interact with each other, and can affect the pathogenesis of psoriasis directly and/or indirectly. The reviewer classified some risk factors according to age or sex. However, these risk factors can affect the pathogenesis of psoriasis regardless of age or sex. As for childhood psoriasis, some risk factors, such as streptococcal infections and vaccinations, might be implicated. However, streptococcal infections can occur at any age. Influenza vaccination is recommended at any age. BCG vaccination is generally scheduled during infancy. We cannot clearly classify risk factors according to age or sex, although we can observe the tendency from the past studies. Some factors were not significantly associated with the development of psoriasis in some past meta-analyses, although those are well established and undoubtedly familiar to clinicians. This is because the impacts of the patients’ age and lifestyle vary among individuals as the reviewer pointed out. Based on your comments, we have incorporated the following sentences in the conclusion paragraph; However, various factors interact with each other and can affect the pathogenesis of psoriasis directly and/or indirectly. For example, obesity, dyslipidemia, and hypertension are associated with the course of psoriasis and are also dependent on the patients’ age, lifestyle, and concomitant diseases. Moreover, the impacts of the patients’ age, lifestyle, and concomitant diseases vary among individuals.
Point 2: Having reviewed the intrinsic risk factors for psoriasis presented in the paper, one may get the impression that there are no relationships between them. I cannot agree with such an opinion or the Authors’ approach towards the problem analysed because obesity, lipid abnormalities and hypertension jointly occur in a number of diseases, including diabetes, CVD and psoriasis.
The approach proposed does not take into account the concomitance, “overlap” of risk factors. The relationships between the extrinsic and intrinsic risk factors cannot be ignored.
I sustain my initial remark that the paper lacks clear presentation of risk factors (e.g. in tables) in the context of age, sex and selected medications. Please note that obesity, lipid abnormalities and hypertension are associated with the course of psoriasis and are also dependent on the patients’ age, lifestyle and concomitant diseases. Separate analysis for individual factors in chronic diseases is not the right approach towards the problem presented.
Conclusions have been modified and better summarise the problem in question.
Response 2: We thank the reviewer for pointing this issue and your suggestion. Based on your comments, we have incorporated the following sentences in the conclusion paragraph; However, various factors interact with each other and can affect the pathogenesis of psoriasis directly and/or indirectly. For example, obesity, dyslipidemia, and hypertension are associated with the course of psoriasis and are also dependent on the patients’ age, lifestyle, and concomitant diseases. Moreover, the impacts of the patients’ age, lifestyle, and concomitant diseases vary among individuals.
Point 3: Additionally, please note that the attempt to identify risk factors needs to be accompanied with their estimation and determination of their OR or HR values. The paper lacks a concise and hierarchical presentation of the parameters analysed. Thus, it does not indicate which of the parameters - based on the data/literature review - may be involved in the etiopathogenesis of psoriasis.
Again, I believe that in terms of the information itself and the way it has been presented the paper does not contribute anything new for clinicians or researchers studying psoriasis. I would like authors to reshape the body of the article.
Response 3: We thank the reviewer for pointing this issue and your suggestion. In the new manuscript, we have revised our manuscript as the reviewer suggests. We have deleted the descriptions about the OR and HR values, because these were rather unclear to explain the risk factors for psoriasis.
Reviewer 5 Report
I assessed that this version is acceptable.
Author Response
I assessed that this version is acceptable.
Response: Thank you very much for favorably reviewing our manuscript.

Round 3
Reviewer 4 Report
The Authors have followed my suggestions, including those pertaining to the values of hazard rates (OR, HR). The correction introduced inproved the clarity of the information presented.
As the type of the manuscript was changed to “review” and the Reviewers’ remarks were taken into account, I recommend the paper for publication. Yet, I still believe that this article does not constitute a new and obligatory reading for clinicians and researchers studying psoriasis.
However, due to the Authors’ considerable involvement in introducing the final corrections, I am willing to accept the paper.
Summary: I recommend the paper for publication.